# Applications of Shaped-Charge Learning

**DOI:** 10.3390/e25111496

**Published:** 2023-10-30

**Authors:** Boris Galitsky

**Affiliations:** Knowledge-Trail, Los Banos, CA 93635, USA; bgalitsky@hotmail.com

**Keywords:** machine learning support for human–machine teams, deep and nearest neighbor learning, structural entropy production

## Abstract

It is well known that deep learning (DNN) has strong limitations due to a lack of explainability and weak defense against possible adversarial attacks. These attacks would be a concern for autonomous teams producing a state of high entropy for the team’s structure. In our first article for this Special Issue, we propose a *meta-learning/DNN* → *kNN* architecture that overcomes these limitations by integrating deep learning with explainable nearest neighbor learning (kNN). This architecture is named “shaped charge”. The focus of the current article is the empirical validation of “shaped charge”. We evaluate the proposed architecture for summarization, question answering, and content creation tasks and observe a significant improvement in performance along with enhanced usability by team members. We observe a substantial improvement in question answering accuracy and also the truthfulness of the generated content due to the application of the shaped-charge learning approach.

## 1. Introduction

Although there have been significant advancements in AI systems that can mimic human language, creating agents that can effectively use natural language (NL) to communicate with humans in productive environments remains a major challenge. The ultimate goal for artificial intelligence (AI) is to develop machines that can converse, interact, plan, agree, and disagree with humans using NL. While progress has been made in language models that can imitate human language, successful collaborative agents must be able to comprehend and communicate their beliefs, goals, and intentions, as well as those of their peers, and plan joint actions that interdependently consider their peers’ objectives and intentions. To support such collaboration, the machine learning (ML) engines used by these agents must combine the efficiency of deep learning networks (DNNs) with the transparency of classical AI methods, including deductive reasoning and deterministic inductive learning.

In the first article for this Special Issue (“Shaped-Charge Learning Architecture for the Man–Machine Teams” [1], we built a neuro-symbolic ML framework that began with neural gradual descent (GD) and concluded with nearest neighbor (kNN) learning. The aim was to merge the benefits of both methods: the accuracy of neural learning coupled with the explainability and meaningfulness of nearest neighbor learning and reasoning, which can confirm and rectify deep neural network (DNN) outcomes. When making a prediction, we first implemented neural gradual descent and then employed nearest neighbors on the resulting candidate prediction. As a result, the prediction process of our shaped-charge ML model reached a climactic phase in which a collection or series of pertinent examples was employed.

One implementation of shaped-charge ML is question answering based on a neural machine learning subsystem followed by a syntactic match with a candidate answer, which is verified by the process and corrected when needed. A similar approach is used for summarization: for each summary phrase, a kNN verifies whether there is a similar sentence in the text being summarized. Another instance of this architecture is transformer-based content generation that is verified and corrected by a web mining fact-checking (kNN-based) component.

Despite the great success of DNN systems like ChatGPT, which have rapidly gained strong interest among the general public for their adequate responses across multiple text genres, there remains an issue with truthfulness. Models such as Stable Diffusion [2] and DALL-E 2 [3] can produce original content in response to queries in the form of social media posts, poems, and even software code, but these models still need verification and possibly correction.

In the first article in a series of two [1], we extended a DNN with a kNN for explainability and accuracy verification. We covered *DNN* → *kNN* with meta-learning, which controls both the learning and prediction processes and also provides the overall system with reasoning capabilities. In this article, for application in text domains, we implemented meta-learning in the form of textual discourse. Syntax and semantics together with a DNN’s recognition of syntax and semantics constitute the object level (i.e., the structure), while the discourse forms the meta level, controlling the object level and its performance.

The contribution of this paper is an empirical validation of shaped-charge learning in several application domains. The study evaluates this approach within the realms of natural language processing (NLP), specifically focusing on summarization, question answering, and content generation. Notably, this evaluation takes into consideration entropy measures inherent to the machine learning system. The findings demonstrate that shaped-charge learning outperforms standalone DNN and kNN in terms of recognition accuracy. Furthermore, it highlights that the superior recognition achieved through shaped-charge learning is explainable within the framework of traditional logical AI principles.

In our application’s domains, kNN works as a structural similarity assessment. A similarity between two sentences is defined as the cardinality of the maximal common subgraph (i.e., syntactic and semantic subgraphs between two sentences). For summarization, it is the similarity of a sentence in the abstract and a paragraph of the text we are summarizing. In question answering, it is the similarity between a question and its answer. For content generation, it is the similarity between a generated sentence and an identified true sentence.

The role of online meta-learning in these domains is different. In summarization, it controls the criteria for finding candidates to be included into the summary. In question answering, meta-learning verifies if the DNN candidate is appropriate and whether a structural similarity assessment needs to be initiated. In content generation, meta-learning issues a request for evidence, decides if a given substitution proposal is plausible, and whether another round of DNN content generation is required.

The remainder of this paper is as follows. In the rest of the Introduction (Section 1), we describe the kNN entropy estimator, followed by meta-learning. Then, we perform an evaluation of the shaped-charge architecture in conducting question answering (Section 2), summarization (Section 3), and content generation (Section 4). In each of these sections, we take a DNN system as the baseline and improve it by using a kNN under the control of meta-learning. The contribution of each component is assessed in the second and third application domains.

Once we demonstrate an increase in the performance of each domain, we proceed to explaining how the advantages of shaped-charge learning are beneficial for functioning as a team. Finally, we formulate an entropy-based foundation for the gradient descent approach connecting DNN and kNN. We conclude by enumerating the strong and weak points of shaped-charge learning architecture.

## 2. Technical Background

Low structural entropy production (SEP) is preferred for optimal performance, setting the stage for maximum entropy production (MEP). With MEP, the steady state of open thermodynamic systems with sufficient degrees of freedom are maintained in a state in which the production of entropy is maximized given the constraints of the system. We approximate a machine learning system with a thermodynamic one and also hypothesize that MEP holds in our case.

MEP is a widely used approach that seeks to determine the most probable distribution functions for observables in statistical systems. This is achieved by maximizing entropy while considering the relevant constraints. Initially applied in ergodic and Markovian systems within statistical mechanics, information theory, and statistics, MEP has been extensively employed across various fields [4]. However, the question of whether the maximum entropy principle can be meaningfully extended to nonextensive, nonergodic, and complex statistical systems and processes has sparked ongoing debate for several decades. In this paper, we explored the application of the MEP principle to such nonergodic complex system as an ML engine supporting a team.

### 2.1. Nearest-Neighbor-Based Entropy Estimator

The differential entropy of a continuous-valued random variable, X, is defined as
(1)$$H\left(X\right)=-\int p\left(x\right)\log p\left(x\right)dx$$
where p(x) is the probability density function of X. Entropy has been an important numerical quantity in ML and statistics, providing a summary measurement of the degree of uncertainty of a system, as well as being related to other important information theoretic measures such as divergence [5] and mutual information [6].

In real-world ML, it is common that the underlying probability density function is not available and only a set of samples is observed. This raises the research problem of estimating the entropy from the observed data only; in particular, flexible estimation approaches that do not assume that the distribution lies in a particular parametric model family are known as nonparametric entropy estimations. kNN estimators [7] and the kernel density estimator [8] have been proposed.

Assume there are *N* independent and identically distributed D-dimensional samples *x*_1_, …, *x_N_*~*P*, where the probability density function *p* is unknown. The classical kNN estimator is written as
(2)$$H\left(X\right)\approx -\frac{1}{N}\sum_{i=1}^{N}\log p(x_{i})\;\approx \;\psi \left(N\right)-\psi \left(k\right)+log\left(c_{D}\right)+\frac{D}{N}\sum_{i=1}^{N}{log\varepsilon }_{i}\;$$
where *ψ* is the digamma function (derivative of (n − 1)!), *ε_i_* denotes the Euclidean distance from *x_i_* to its nearest neighbor, $c_{D\;}=\;\frac{\pi^{\frac{D}{2}}}{\Gamma (1+\frac{D}{2})}$, and *ψ*(*N*) − *ψ*(*k*) is the correction term.

The classical estimator has been widely applied in many different research problems [9]. It has also been adopted to nonparametrically estimate mutual information. Lu and Peltonen [10] introduced a novel entropy estimation method called the kNN estimator with ellipsoidal correction that is directly applicable both to entropy estimation and to mutual information estimation. The idea is to construct a local ellipsoid instead of an ε-ball, so that the insides of the local ellipsoid samples can be assumed to be more uniformly distributed and, hence, will better fit the uniformity assumption of kNN-based entropy estimations.

### 2.2. Meta-Learning

Meta-learning refers to a subfield of ML of which the focus is on developing algorithms and models that can learn how to learn. In traditional ML, a model is trained on a specific task, and its performance on that task depends on the quality and quantity of the training data. However, in meta-learning, the goal is to enable a model to acquire knowledge or adapt quickly to new tasks with limited data.

Meta-learning involves training a model on a variety of related tasks or datasets so that it can learn common patterns, strategies, or representations that can be applied to new, unseen tasks. The key idea is to make the learning process more generalizable and transferable so that a model can leverage its prior experience to rapidly adapt to novel tasks.

Meta-learning has applications in various domains, including few-shot learning (where models are trained to make accurate predictions with very few examples), reinforcement learning (where agents learn how to adapt to different environments), and optimization (where models learn to optimize their own learning process). It is a promising area of research for improving the efficiency and adaptability of ML algorithms.

## 3. Question Answering

Despite the impressive achievements of deep neural networks in question answering, they still fail to accurately answer a considerable number of simple questions [11,12]. Multichoice question answering (Q/A) is a difficult challenge that requires the selection of the correct answer (A) from a set of options when given a passage and a question (Q). This is especially true when dealing with ambiguous semantic roles, multiple instances of the same semantic type, or complex syntactic constructions. To address this issue, minimizing the structural entropy (SEP), techniques such as the direct assessment of the similarity between a Q and an A, and using the maximal common substructure of the linguistic representations [13] can be beneficial.

Answering factoid questions has been a difficult endeavor for industrial conversational interfaces. Current systems are typically trained with a collection of questions and answers prepared by business users. However, since large enterprises like banks offer a wide range of products and services, an extensive number of questions must be trained to ensure coverage is sufficient. Additionally, because of continual updates to products, conditions, and regulations, maintaining an industrial training dataset is difficult, making it challenging to create a search engine and content management system for a customer relationship management (CRM) portal [14]. To address this, extracting knowledge from sources such as syntactic parse trees and semantic abstract meaning representation (AMR) parse results can offer a systematic method of answering attribute-value questions [15]. By traversing both the syntactic and semantic parses in tandem, this approach allows for complete coverage of potential question phrasings given an answer.

To enable computers to “truly understand” the meaning of a sentence, semantic parsing is applied to map a natural language sentence to a semantic representation. One such semantic representation is called an AMR, which is represented as a rooted, directed, acyclic graph with labels on edges (relations) and leaves (entities). AMR involves a bank of English sentences and their logical meanings, making it easier for semantic parsers to be used. Existing today, a corpus of over thirty thousand sentences with AMR representation [16] is available. To compute the meaning of a sentence (MEP), it is crucial to understand the entities and relations between them (SEP). Leveraging efficient AMR parsers [17,18,19], this work implemented an instance of a kNN approach, which is termed a “generalization of AMR” (AMRG).

Deep neural networks (DNNs) can answer a large number of arbitrarily-phrased questions. A deterministic technique such as AMRG can be used to answer the remaining questions. This involves verifying the correctness of the DNN’s answer, taking into consideration the syntactic, named entity relations (NERs) and semantic role information.

We hypothesized that combining a DNN system and a direct syntactic/semantic similarity approach could be beneficial, since they rely on different feature spaces [20]. DNN-based approaches often fail because of a lack of an online phrase structure similar to the one available in the training set or an inability to generalize to cover various cases. In the case of ChatGPT, it frequently hallucinates when there is a lack of similar enough examples in the training set [21]. To explore this idea, we analyzed how AMR-based matching and state-of-the-art DNN-based approaches can be integrated together to form a combination of two classifiers for candidate answers.

The architecture of this graph representation matching for Q/A is illustrated in Figure 1. Both the question and text from which an answer is to be extracted are subject to both syntactic and semantic parsing to construct an AMR. Additionally, other tagging pipelines are used, such as named entity recognition (NER), sentiment, and emotion.

At the next step, all available representations for Q are aligned with each other, and all representations for the answer text (context) are aligned with each other as well. Finally, a search for the answer is an alignment of a hybrid (aligned) representation for the Q against that of the A [14]. An answer fragment is a maximal common subgraph between the aligned Q and the aligned A [13].

We consider a DNN-based approach as an end-to-end “black-box” system. We took a state-of-the-art approach to the Q/A task based on the ALBERT transformer model (see [22]). We used the same (single) model as was built, trained, and provided by the authors.

We evaluated the AMR-based reading comprehension with the following settings:(1)Standalone Q/A;(2)Error correction of the DNN’s Q/A, whereby each question is handled by the DNN and then the AMR attempts to verify/correct it;(3)Hybrid approach in which both the DNN and AMRG produce answers, and a meta-agent then decides on which answer to choose.

In the hybrid approach (Figure 2), the DNN module acts as a traditional neural MRC in the first step. Subsequently, the AMRG meta-agent with the architecture outlined above checks if the answer is linguistically and semantically correct, substituting it into the question and performing syntactic and semantic matching with the answer text (context). A “good match” means low divergence, low information gain, and the highest entropy output.

For comparison purposes, we also evaluated a search of the same queries through the entirety of the available documents. In addition to the search F1 measure, we used the normalized discounted cumulative gain (NDCG) measure for the top five search results (Table 1).

A conventional TF*IDF-based search was integrated with an MRC component to implement a full corpus search. Lucene search was used to identify a set of candidate documents, and MRC was applied to each one.

We used the Stanford Question Answering Dataset (SQuAD) [11] for evaluation. It is a reading comprehension dataset with more than 100,000 question–answer pairs based on 500+ Wikipedia articles. Every question is answered with a segment of text, also known as a span, from the related passage. SQuAD was created by having people compose questions for a given Wikipedia article and then select the answer span. It has more data than previously existing reading comprehension datasets.

We evaluated our system on both versions of the SQuAD dataset: 1.1 and 2.0. A DNN can typically find the answer to a question in a short text, but it struggles on questions for which the answer is not explicitly stated in the context (high SEP). Rajpurkar et al. [11] trained their system on SQuAD 1.1 and tested it on unseen questions, so in version 2.0, additional unanswerable questions that appear similar to answerable ones were added. We used test subsets for both versions of the dataset for our evaluation.

For many questions and answers, especially those beginning with “Wh-” and “How to”, there is significant syntactic similarity. When a Wh-word is substituted with a placeholder of a certain semantic type (like a noun entity), the similarity between the Q and A can easily be identified.

MRC models have been known to have a decreased performance (a low MEP) when answering long questions due to the difficulty of processing extensive texts with a neural model and distinguishing among text components [23]. To improve this, research has been conducted [24,25] suggesting that neural networks could benefit from using syntactic information instead of disregarding it. Errors made by a syntax-free neural network MRC can often be attributed to confusion with prepositional phrase attachment, and it has been found that providing a high-quality syntactic and semantic parse can lead to considerable advantages, even when constrained inference with a low-accuracy predicted parse is used.

Zhang et al. [23] showed a 4% boost in the MRC’s performance on the SQuAD 2.0 dataset, increasing the F1 score from 83.6% to 87.9%, by incorporating semantic information into BERT. To achieve this, they developed SemBERT, an improved language representation model based on a BERT backbone, which can incorporate contextual semantics from a pretrained semantic role labeling.

Using syntax to guide text modeling and to improve linguistically motivated word representations, Zheng et al. proposed SG-Net, which was applied to the pretrained language model BERT. The explicit syntactic constraints incorporated into the attention mechanism of an SG-Net resulted in a considerable performance boost (maximum entropy), with F1 scores reaching up to 87.2 on SQuAD 2.0. This demonstrates the necessity of effectively modeling the linguistic knowledge from lengthy passages and eliminating noise for improved machine reading comprehension (least SEP).

In 2018, Hu et al. proposed a novel “read + verify” system that utilizes an answer verifier to determine if the proposed answer is supported by the input snippets, similar to the present study. The authors added two extra losses to aid the reader in handling answer extraction, as well as no-answer detection, and studied three various structures for the answer verifier. Their experiments on the SQuAD 2.0 dataset yielded an F1 score of 74.2 on the test set, representing a state-of-the-art result at that time.

Table 1 shows the performance of our standalone AMRG system using both the SQuAD 1.1 and 2.0 test data for the evaluation. We observed a 5–6% increase in the accuracy when going from a system that only has a syntactic level to one with only a semantic level (without any alignment). This increase means a lower structural SEP, low divergence, high information gain, and higher SEP [26]. Further, the joint syntactic and semantic system with alignment delivered a 9–10% improvement in the accuracy, as indicated in the bolded row. Despite this, our system is still 8% below that of a state-of-the-art DNN system, as seen in the bottom row. This table is valuable for our ablation study of the mutual roles of an AMR and a variety of DNN architectures for Q/A. In the top three (shared) rows, we show three settings for increasing the complexity of the kNN family, which were implemented as syntactic and semantic generalizations. In the bottom five rows, we show the accuracy of a spectrum of DNN settings.

The ALBERT transformer model, introduced by Lan et al. [22], excels in parameter efficiency, significantly reducing the model’s size while maintaining or even improving performance, making it more computationally and memory efficient. ALBERT’s innovative cross-layer parameter sharing enhances model regularization and generalization. It introduces the sentence order prediction task, a more linguistically meaningful objective compared to BERT’s next sentence prediction, leading to better contextual embeddings. Furthermore, ALBERT demonstrates improved handling of longer documents and exhibits superior transfer learning capabilities, outperforming BERT on various tasks while being scalable to different resource constraints.

**Table 1 entropy-25-01496-t001:** Performance of the standalone AMRG.

	SQuAD 1.1			SQuAD 2.0		
	P	R	F1	P	R	F1
Syntax trees only	71.6	72.9	72.2	67.2	68.3	67.75
AMR representation only	75.4	78.3	76.8	74.2	73.5	73.85
Aligned syntax trees and AMR representation	85.3	86.3	85.8	83.4	84.1	83.75
DNN [23]						87.9
DNN (SG-Net) [23]						87.9
DNN with syntactic analysis “read + verify” [27]						74.2
DNN (ALBERT) [22]			94.8			90.9
DNN ensemble (ALBERT + DCMN) [22]			95.2			92.2

We can observe in Table 2 that the preferred Q/A setting is an answer provided first by the DNN and then error correction by the AMRG as the second step. This optimal configuration is (the three rows on the top with the third row bolded):(1)Approximately 2% above a standalone DNN approach.(2)A meta-learning approach, which takes the DNN score and maximal common subgraph score to decide which answer to choose, was used in this configuration. However, as this configuration is not optimal, it is not described in detail here. This configuration was approximately 9% higher than the hybrid standalone AMRG and standalone DNN;(3)Approximately 9% above the standalone AMRG.

The two rows at the bottom provide further details on the optimal hybrid system. The false-positive rate of the AMRG component is noticeable, resulting in some correct answers being incorrectly classified as incorrect and the wrong answer being delivered. However, this happens less often than the opposite, so the overall impact of the AMR component is positive, i.e., more answers are turned from incorrect to correct than vice versa. This occurs since the nature and sources of information for an AMRG Q/A are totally different from those of a neural, ML-based Q/A. The error-correction rate is lower than the overall performance because the DNN makes the most mistakes in complicated cases.

We present the relative kNN entropy estimate in the right column, as described in Section 2.1. We observed a very modest 1.89 ± 0.12% improvement in the performance due to the DNN + AMRG architecture. The AMR-only row is a useful case for our ablation study.

AMR parsing is conducted using a neural process, which is more error prone than MRC learning. Nevertheless, because of the correspondence with a dependable syntactic parse, the representation created by AMR functions as a trustworthy basis for our MRC. One can observe the role of a meta-learning agent deciding which prediction scenario to apply in a given case (fourth row in Table 2 and Table 3).

Hence, the boost in the performance for the full-index search (the third row in comparison with the first row) was 14.5 ± 4.2%, as measured by NDCG @ 5 hits.

To further evaluate the effectiveness of our search system, we conducted a full-corpus search of Wikipedia pages. Table 3 highlights how a combination of a DNN and AMR yields enhanced performance over hybrid and standalone systems. Both the F1 and NDCG measures show a marked improvement in the accuracy of the first search result, as well as the top five search results. The performance of the information retrieval (IR) component that finds the candidate documents had a negligible impact on the overall Q/A performance. Our performance in terms of NDCG is shown by ranking five search results.

We performed a case study on what kinds of errors are produced with each method. It turns out that the kinds of errors are different:An AMRG performs incorrectly when syntactic similarity is low or when important phrases form a longer span in the sentence;A DNN experiences difficulties forming the meaning of multiword expressions.

An analysis of the intersection between the errors of the AMRG and DNN reveals a 35% overlap.

## 4. Text Summarization

Substantial progress has been made with DNN-based methods; however, generating coherent and informative abstract summaries is still a challenging task. A human-like reading strategy has not been significantly leveraged in abstractive text summarization approaches, yet it is capable of improving the effectiveness of a summarization by relying on the process of reading comprehension and logical thinking. Motivated by the human-like reading strategy that follows a hierarchical routine, [28] designed a hybrid learning approach for abstractive text summarization. Their model consisted of three major components: a knowledge-based attention network; a multitask encoder–decoder network; and a generative adversarial network believed to match the different stages of the reading strategy of human team members.

Text summarization can have a similar hybrid architecture as question answering. A summarization obtained using deep learning can be generalized with the text to assess the quality of the summarization based on the syntactic features of the generalization, with the result as follows:If a summary is a subsentence of one of the sentences in the original text, as determined by the generalization result, usually it is not a very creative summary;If a summary sentence is a rephrasing of several consecutive sentences in the text, as determined by a generalization, then this summary usually makes good sense;If a summary is a rephrasing of multiple original sentences combined together, the chance is it is of a higher quality (higher MEP);If the generalization operation verifies the continuity of the process, the generalization operation computes the least general generalization of the syntactic and/or semantic representations, implementing the kNN.

Unlike in the case of a search, the generalization operation cannot produce a summary on its own; it can only verify its syntactic and semantic quality and consistency.

Cachola et al. [29] proposed a new form of extreme summarization for scientific papers based on a high source compression that requires expert background knowledge and understanding of the complex domain-specific language entailed. The summarization exploits titles as an auxiliary training signal, improving the baselines under both automated metrics and human evaluations.

In the DNN abstractive summarization field, conventional transformer-based models are often limited to summarizing the wrong part of the document with respect to the main topic. To tackle this problem, Gao et al. [30] came up with the task of a reader-aware abstractive summary generation that leverages reader comments to assist the system with yielding better summaries of the main topic. Makino et al. [31] designed a global optimization method with a constraint on the length of the summary for DNN summarization models, boosting the probability of generating summaries that have high evaluation scores and set within a desired length.

A Weibo dataset [30] consists of document–summary–comments triple data from Weibo, which is the largest social network website in China. Users can read a document and post a comment about the document on this website. Each sample of data contains a document, a summary, and several reader comments. Most comments are about the readers’ opinions on a focused aspect of the document. In total, the training dataset contains 0.86 million training samples. The average length of a document is approximately seventy words, and the average number of comments in a document is almost ten.

SciTLDR [29] is a multitarget dataset of 5000 TLDRs (“too long; didn’t read”) over 3000 scientific papers in the computer science domain. The training set contains 2 thousand papers, each with a single true TLDR. The development and test sets contain approximately 600 papers each, with 1.5 thousand and 2 thousand TLDRs, respectively. The CNN/Daily Mail Corpus [32] is extensively employed in abstractive text summarization. The dataset includes news from CNN/Daily Mail associated with multi-sentence summaries generated by human team members. In total, it consists of 200,000 training instances; 13,000 validation instances; and 11,000 test instances. There are almost a thousand words on average for an article and 56 tokens on average for a summary.

We show the results of four DNN summarization systems for the ROUGE measure and the improvement in the MEP, whereby the kNN step rejects a given summary (Table 4). The summarization improvement can also be characterized as a boost in output entropy productivity. The MEP productivity is contingent on the structure, i.e., the maximum MEP is not a guarantee but a maximum allowed by the structure, when all else is fixed. For example, a current summary without a semantic rephrasing can be at MEP, but its quality is still below a summary when semantic rephrasing is applied. In the right column, we show the kNN entropy drop achieved by the summarization improvement. kNN entropy’s calculation is presented in Section 2.1.

In the case of rejection, a document title is given instead. We measured the improvement using ROUGE-L (longest common subsequence). One can see that the improvement was the lowest in the scientific domain and the largest in the more conventional texts such as provided by CNN. Although sometimes the generalization operation rejected the correct summarization because it deviated from the original text too much, the overall impact was positive. We conclude that the generalization operation helps to filter out an irrelevant summary as per the ROUGE-L measure.

We observed no significant improvement (0.7 ± 1.2) in the case of the scientific domain and a minimum boost of up to 3.5% in the CNN domain considering the deviation in the ROUGE-L calculation.

## 5. Open Domain Text Generation

Large language models (LLMs), such as GPT-3 [33] and ChatGPT, have exhibited exceptional proficiency in producing coherent, informative, and fluent natural language texts. This is often attributed to the copious amount of world knowledge that these models contain, allowing them to generalize and apply this knowledge effectively. Nonetheless, the encoding of knowledge in LLMs is imperfect, resulting in some loss, and the process of generalization may also cause some “memory distortion” [34]. Consequently, these models have a tendency to generate hallucinations, leading to incorrect information, which can be detrimental when utilized for tasks that are critical to mission success.

The main goal of this section is to identify hallucinations in content generated by a DNN-based system and repair these hallucinations with factual content taken from various sources. We used the structure and content flow from the raw text, replacing each hallucinated phrase with one composed of factual information gathered from the Web. This enabled us to create original content produced by a DNN with a factual truthfulness provided by the Web or other reliable sources of information.

Whereas the role of a DNN is to produce candidate content (possibly with hallucinations), a kNN verifies this content with respect to factual accuracy. This is performed by finding a “neighbor” for each phrase or sentence that is a text available on the Web or elsewhere that can be trusted. There is no scalability limit for this kNN operation, as web content is available to support/accept/reject an arbitrary phrase or sentence. If such a support is not found, we assume that this sentence is factually wrong and is, therefore, rejected and assigned for value substitution. A kNN can be implemented with direct structural similarity measurement and no attention mechanism is necessary.

A kNN operates on a structural representation of sentences, unlike the DNN attention mechanism. A kNN takes a syntactic and semantic representation of a generated sentence and matches it via graph mapping algorithms to syntactic and semantic representations of a true sentence identified with meta-learning on trusted websites or other authoritative sources of information. This procedure will be described in detail in this section. Hence, the correction of DNN content is domain- and language-independent, if syntactic, semantic, and web search components can support a given language.

For our evaluation, we selected web sites that were indexed and rated highly by major web search engines as sources of true information to perform our fact-checking of DNN-generated content. Although factual errors are still possible, they occur approximately a thousand times less frequently than hallucinations in DNN content. Misinformation in popular portals indexed by Google and Bing and shown in top results is very rare. Only social media misinformation created by humans and hallucination rates are comparable [35]. The reader can obtain an estimate for the DNN hallucination rate from the evaluation data in this section. 

Zhang et al. [36] presented a DNN system tailored to the health domain of personalized recommendations, such as drug intake. This system generates text from a structured dataset. To obtain raw text, an arbitrary DNN content generation system, such as GPT-4, can be used with a seed sentence as input. We now describe our architecture for a hallucination correction system.

The DNN subsystem consists of three components:(1)A data collection module;(2)A content plan generation module;(3)A description generation module.

In the data collection module, [36] gathered a dataset of triples of attributes, a content plan, and an entity description. They extract the attributes by querying Wikidata for Resource Description Framework (RDF) triples that have the target entity as the subject. The description is taken from Wikipedia by extracting the first sentence of the Wikipedia page of a target entity. The content plan is generated by finding the order of the attributes in the description through string matching. In the content plan generation module, a pointer network is used to learn a content plan that helps the attention model of the description generation module arrange the attributes in the right order. This module consists of four components:(1)An attribute encoder that encodes a set of attributes into a vector by computing the average of the linear transformation of each token embeddings of the attributes;(2)A pointer generator that generates a sequence of indexes (i.e., pointers) that represents the order of the attributes in the description;(3)A content plan generator that generates the content plan based on the learned pointers;(4)A content plan encoder that encodes the learned content plan to be used in the description generation module.

In description of [36], generation module, the content plan is integrated into the encoder–decoder model by employing the content plan-based bag-of-tokens attention model. The content plan is encoded using a long short-term memory system to capture the links between the attributes, and the coverage mechanism [37] is adapted to track the order of the attributes in the content plan for computing the attention of the attributes. This attention model selects the salient attributes conditioned by the content plan, thus providing a better context for each decoding time step.

For each sentence in the raw text, we performed a deterministic fact-checking process as an implementation of the kNN learning. We iterated through each sentence and made as few modifications to the syntactic structure as possible. We first used syntactic criteria to determine if the sentence should be retained in the corrected content. We then proceeded to fact-checking to form a family of queries from the sentence to obtain the true candidate sentences. These true sentences were extracted from the search result snippets and identified documents, and then they were matched against the given raw sentences to determine the most optimal substitution. Syntactic and semantic alignments were built and the entities and phrases to be substituted were identified. The architecture chart presented in the end of this section shows the major decisions: whether a raw sentence can be modified; if yes, should it be an entity substitution or a phrase substitution? The structure of coreferences in the raw text must be taken into account when making substitutions for multiple sentences. The discourse structure of the resulting text should then be compared to that of the original text [1].

We looked at an example of a raw and accurate choice for a person suffering from Liddle’s syndrome [38]. This hereditary condition is characterized by early and often serious hypertension related to low plasma renin activity, metabolic alkalosis, low blood potassium, and normal to low amounts of aldosterone.

A woman with Liddle’s syndrome presented with severe symptomatic hypokalemia. Her doctor reasoned as follows [raw text]:-She has potassium depletion;-Spironolactone is a potassium-sparing drug;-Spironolactone will cause her to retain potassium;-Her serum potassium concentration will normalize.

She took a full dose of spironolactone for several days, based on this logical reasoning, but still had severe hypokalemia.

Her doctor should have reasoned as follows [true text]:-She has potassium depletion due to Liddle’s syndrome, a channelopathy that affects epithelial sodium channels;-There is a choice of potassium-sparing drugs;-Spironolactone acts via aldosterone receptors, amiloride, and triamterene via sodium channels;-In Liddle’s syndrome, an action via sodium channels is required.

When she was given amiloride instead of spironolactone, her serum potassium concentration quickly returned to a normal level. This illustrates the significance of comprehending the nexus between the physiology of a condition and how a medicine works. Specifically, channelopathies are diseases caused by a disruption in the operation of ion channel subunits or the proteins that control them. These illnesses can either be hereditary (i.e., generally caused by a mutation or mutations in the genes that encode them) or acquired (i.e., frequently caused by an autoimmune response against an ion channel).

We attempted to use generalization/alignment to compare raw and true texts. Alignment of a graph to a graph is a process that retains the labels of links between the source and target nodes of the two graphs. We created an alignment between two semantic graphs and their corresponding syntactic trees; this alignment respects the semantic and syntactic connections between the source and destination graphs.

In Figure 3 below, the green arrows show that the alignment maps potassium depletion to itself, potassium-sparing to itself, and raw mental action to true mental action. The red arrows show that the alignment substitutes “raw retain potassium” for the “true sodium channel”.

We start with another example of a raw text and its correction (Figure 4). We demonstrate that there are multiple inconsistencies (marked by red arrows) in the raw sentence that needed to be addressed. We provided the raw sentence and the correct sentence. The incorrect values (which differ from the correct values) in the raw sentence are marked in red, while the true sentence is highlighted in yellow. Additionally, the values that are updated in the true sentence are marked in blue, including the relationships and attributes that are confirmed by the true sentence.

The top six sentences can be successfully corrected to yield meaningful factoid sentences, preserving relationships and attribute types while taking the values from the true sentences. The bottom two raw sentences, however, do not yield meaningful factoid sentences and should be ignored.

Hence, from what we can observe, we can formulate the following correction rules, reducing wasted structural entropy to improve the output:(1)If an individual value can be updated, the sentence is retained.(2)Otherwise, we need to update the syntactic structure of the sentence mined from the Web to match the raw one; so if no associated fact can be identified, the sentence is removed and the closest sentence found in an available source is used. To form the syntactic skeleton of the raw sentence and generalize it with the sentence obtained from the source, some phrases may have to be removed.

Hence, the kNN operates on explicit structures like syntactic and semantic trees. We match one word of a certain part of speech with another word, which can be synonymous or not synonymous. We also match phrases of the same type. At the same time, an attention mechanism matches embeddings of phrases in a way hardly interpretable by a humans. Therefore, humans can perform feature engineering for a kNN.

Our evaluation was in the health domain. We gathered data on drug–condition (disease) pairs from two sources: www.rxlist.com (accessed on 2 June 2021) and www.medicinenet.com (accessed on 5 June 2021). From the former, we selected 3000 drug names and from the latter 500 conditions. We considered the relationships between the drugs and conditions to either be drug-treats-disease, side-effect-of-drug-under-condition, or drug-cause-side-effect-under-condition. We refer to this data as the D–C pair set. The objective was to investigate whether a drug has a side effect with a given condition, might affect it, or might cause complications related to it. This way, we can make personalized drug intake recommendations for a patient with a particular condition.

By exhaustively iterating through all drug–condition pairs and retaining those with an explicit relationship, we were able to compile a list of 4.6 thousand pairs. With this list, we can now experiment with writing personalized content for a patient with condition C taking drug D.

We characterized the correction scenarios in Table 5 (a) for GPT 3 and Table 5 (b) for GPT 4. In the first column on the left, the class of disease symptoms is shown for each assessment group. The averaged relative number and the absolute number of sentences subject to the certain corrections are shown in columns two to four:(1)Replacement of the entity;(2)Replacement of the whole phrase;(3)Rejection of the whole sentence.

The relative number of corrections is computed as a portion of all of the sentences in the raw text. The right three columns show:(1)The absolute and relative number of sentences accepted without modification;(2)The total number of raw sentences;(3)The total number of true sentences mined and used to repair untruthful raw sentences.

On average, it takes 18 sentences mined from the Web to correct a raw drug recommendation text of nine sentences. Typically, three of these sentences are corrected by substituting entities, two by substituting phrases, and more than two are rejected as noncorrectable.

We subjected the corrected content to another round of correction using a different web source to automatically assess the error rate. By counting the number of sentences that fail fact-checking, we can obtain a better understanding of the accuracy of the original correction procedure, although this automated assessment may lead to false positives and false negatives. However, the advantage is that it can be applied to the entire evaluation dataset, and structural (processing) entropy costs are reduced.

The content verification and correction architecture is shown in Figure 5. Two processing pipelines are implemented, combining sequential DNN and kNN processing: direct fact-checking and an iterative loop supplying the LLM with correct evidence. The common processing part (i.e., fact-checking) is shown on the top left. A text obtained from the large language model (LLM) is decomposed and decontextualized into verifiable claims which can potentially introduce an incorrect factual data into text. Then, for each claim, fact-checking is performed according to the following kNN scenario:(1)Form a search query against a trusted source (such as a website or corpus of documents);(2)Run this search query and find search results (nearest neighbors to the claim being fact-checked);(3)Collect the search results and verify if they are consistent with each other using NL the inference. If there is a contradiction, attempt to find a least inconsistent result by applying a logical system of argumentation [1];(4)Once the best evidence is obtained, the system selects the pipeline. If the correction cannot be made inside the system, the evidence is passed back to the LLM; however, there is a risk that LLM will hallucinate again. Otherwise, the system performs the substitution of the entities and values from the kNN result and outputs the corrected text. This decision is made by the meta-learning.

The DNN components in the architecture are shown in blue and logical reasoning in red. The meta-learning’s if/then component is represented by the rhombus shape.

The resultant content generation and correction quality is displayed in Table 6. According to the automated fact-checking, only 6% of the raw texts can be accepted without repair. After correction, 84.1% of texts are free of wrong facts, reducing the number of wrong facts in the raw text from 69.6% to 6.6%.

We measured the frequency at which repaired sentences or phrases were not confirmed with a true sentence or phrase by repeating the truth–repair procedure automatically in order to verify the accuracy of the generated content.

Finally, we performed a more accurate hybrid assessment of the resultant quality of the content correction (Table 7). This combined the automated fact-checking on the Web with a manual assessment of the meaningfulness. We show the assessment result in the second raw text and verify that the deviation between the auto and manual assessments did not exceed 10%.

Table 5 (a) shows the step-by-step improvements achieved by the kNN algorithm. The corrections occur by using entity substitution, phrase substitution, and a rejection of the whole erroneous sentence that cannot be corrected.

This article was written during the great success of ChatGPT and GPT4, which have acquired millions of weekly users. One set of potential applications for *ChatGPT* → *kNN* (fact-checking) is customizing ChatGPT to a specific domain, such as the regulations of a particular business-like bank, with the data coming from averaged banking information from the Chat GPT training corpus. We repeated our earlier evaluation settings but leveraged the full power of GPT4. The results are shown in Table 5 (b).

One can observe that in the health domain, 26% of sGPT-4’s sentences were properly corrected when entity or phrase substitution was performed.

In addition to the syntactic and semantic measure of the kNN distance, we observed how the discourse structure (overall logical organization of a text) is retained in the course of the content correction. The right column in Table 6 shows the measure of how the resultant discourse structure is broken (the sequence of the discourse relationships is altered) under the kNN-based correction procedure. Fourth and fifth column with the rates of wrong facts are greyed.

Table 7 presents the results of the ablation study. Columns two, three, and six contain the data for an arbitrary domain and columns four and five the health domain evaluated in Table 5 and Table 6.

Conducting a manual assessment on top of an automatic one on the truthfulness of corrected phrases led to a 0.6% more accurate detection of cases in which the system correction was still wrong. A reduction in the web mining from the whole Web to a limited number of health-related websites led to the leaking of erroneous texts in 4.1% of cases. A keyword-based instead of full parse tree-based generalization lost 3.4% cases, and by simplifying to phrase-based and entity-based substitutions, the performance only dropped by 2.3 and 1.1%, respectively.

## 6. Discussions and Conclusions

Our findings suggest that augmenting a DNN with a kNN approach and, subsequently, applying meta-learning to this extended model can substantially enhance both the accuracy and interpretability of machine learning, ultimately, improving support for human–machine collaborations. We demonstrated performance enhancements across question answering, summarization, and content generation tasks. In these tasks, we took the DNN results and improved/verified/corrected them with the help of a kNN, with the whole process controlled by meta-learning.

The time complexity of the kNN for a single prediction can be expressed as O(*nd*), where *n* represents the quantity of the training examples, and *d* is the number of features. This complexity arises from the necessity of calculating the distance between each query point and every other point in the dataset. Let us now estimate the complexity of the transformer network. Let X be the input to a self-attention layer. Then, X will have the shape (*n*, *d*), since there are *n* word vectors (corresponding to rows), each of dimension *d*. Computing the output of self-attention requires the following steps in the case of single-headed self-attention:

Linearly transform the rows of X to compute the query *Q*, key *K*, and value *V* matrices, each of which has the shape (*n, d*): this is accomplished by post-multiplying X with three acquired matrices of shape (*d, d*), yielding a computational complexity of O(nd^2^).

Compute the layer output as $Attention\left(Q,K,V\right)=softmax\left(\frac{QK^{T}}{\sqrt{d_{k}}}\right)V\;$over each row: Computing *QK^T^* has complexity O(n^2^d), and post-multiplying the resultant with *V* also has complexity n^2^d. Therefore, the total complexity of the layer is O(n^2^d + nd^2^), which is approximately √nd times higher than a kNN. The same complexity estimate is valid for the fine-tuning step. Therefore, substituting the DNN with a kNN, when possible, has complexity-related benefits. Syntactic and semantic representations are built relying on the DNN, so the complexity estimate described above covers these cases. Since we built these representations for shortlisted candidate sentences rather than the whole text, the complexity does not grow much in our kNN applications for content correction.

As our application domains demonstrated, shaped-charge learning offers several advantages compared to other hybrid DNN + statistical/logical ML architectures:(1)Standalone kNN holds a superior position in terms of explainability, closely aligning with the logical underpinnings of inductive learning. Shaped-charge learning leverages the inherent advantages of kNN, including computational efficiency, interpretability, applicability to both regression and classification, and high accuracy.(2)The sequential progression from DNN to kNN in shaped-charge learning guides the learning process, gradually zeroing in on correct predictions within initially vast search spaces. This step-wise approach effectively addresses the limitations of the standalone DNN and standalone kNN methods.(3)Shaped-charge learning incorporates a meta-learning mechanism that provides precise control over the choice of methods applied to specific data at each decision point. The meta-level support, facilitated using gradient descent in the DNN → kNN transitions, ensures an efficient fusion of manually selected features for the kNN and automated feature engineering through DNN.

The interest in enhancing LLMs with external knowledge is on the [39,40]; however, almost all of the previously suggested techniques necessitate the fine-tuning of the LLM’s parameters. This can be exceedingly costly (high structural entropy cost), particularly as the size of LLMs expands exponentially. Therefore, performing fact-checking outside of the LMM, as we described in this paper, is beneficial.

Browning and LeCunn [41] write that as large LMs become more robust and usable, there seems to be less consensus over how designers and users should understand them. LM-based systems have solved many “common sense” linguistic reasoning benchmarks over the years, many of which promised to be comprehensible only to a machine that “is thinking in the full-bodied sense that is usually reserved for people”. Yet these systems rarely seem to have the common sense promised when the test is broken; moreover, they are still prone to offering nonsense and dangerous advice. This leads to a troubling question: how can these systems be so smart, yet also seem so limited?

Finally, extending a DNN with a kNN and performing meta-learning over this extension significantly improves the overall accuracy and explainability of ML, as well as offering human–machine team support, as was demonstrated with the question answering, summarization, and content generation projects. In the summarization task, we archived a more than 3% improvement in the accuracy. In the question answering task, the increase was greater than 10%. For the content generation, more than 26% of sentences generated by GPT 4 needed corrections and were properly corrected. We demonstrated the role of a kNN and meta-learning in improving the performance of a standalone DNN.

Our main result in the content correction domain is that the shaped-charge framework is suitable for augmenting black-box LLMs (e.g., ChatGPT) along with external knowledge. Whereas in the work of Peng et al. [34], the external knowledge provided as part of the LLM prompt helps to generate a greater number of responses that are more grounded in external knowledge that is relevant to the current conversation, in our proposal, such knowledge is leveraged outside of the LLM. The automated feedback [34] elicits the skill of a “follow-up correction” for ChatGPT to produce revised responses. These responses are scored higher than the default ChatGPT responses according to a given utility function.

The limitation of the conducted research is as follows:(1)We explored the kNN family of explainable ML algorithms; however, structures such as decision trees and formalisms of inductive logic programming [42] can potentially play a similar role.(2)An application of shaped-charge in the text domain was developed. However, we do not have an assessment of its performance in other NLP domains, for example, textual entailment and domains beyond text.

We plan to address these limitations in our future studies.

## Figures and Tables

**Figure 1 entropy-25-01496-f001:**
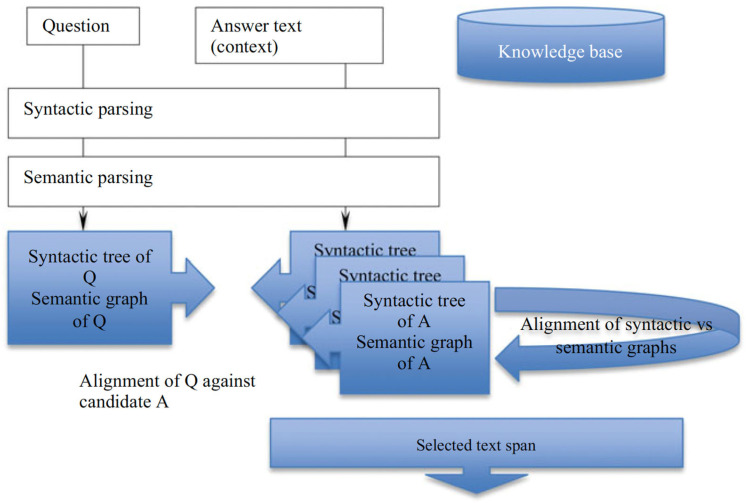
AMRG component architecture for finding an answer to a question with respective structural graph alignment. The correct answer reflects MEP.

**Figure 2 entropy-25-01496-f002:**
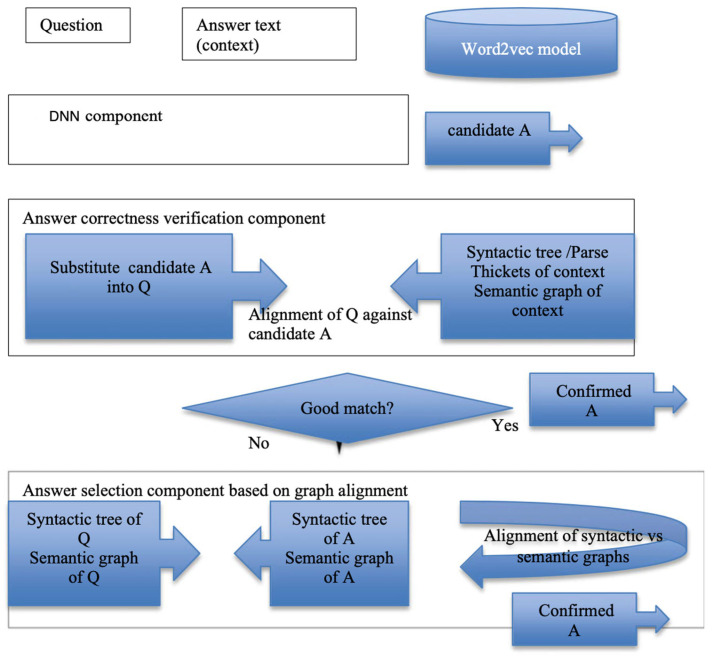
Error identification and answer selection scenario for the hybrid DNN and AMRG system. Meta-learning assesses whether the match is satisfactory and decides upon consecutive steps.

**Figure 3 entropy-25-01496-f003:**
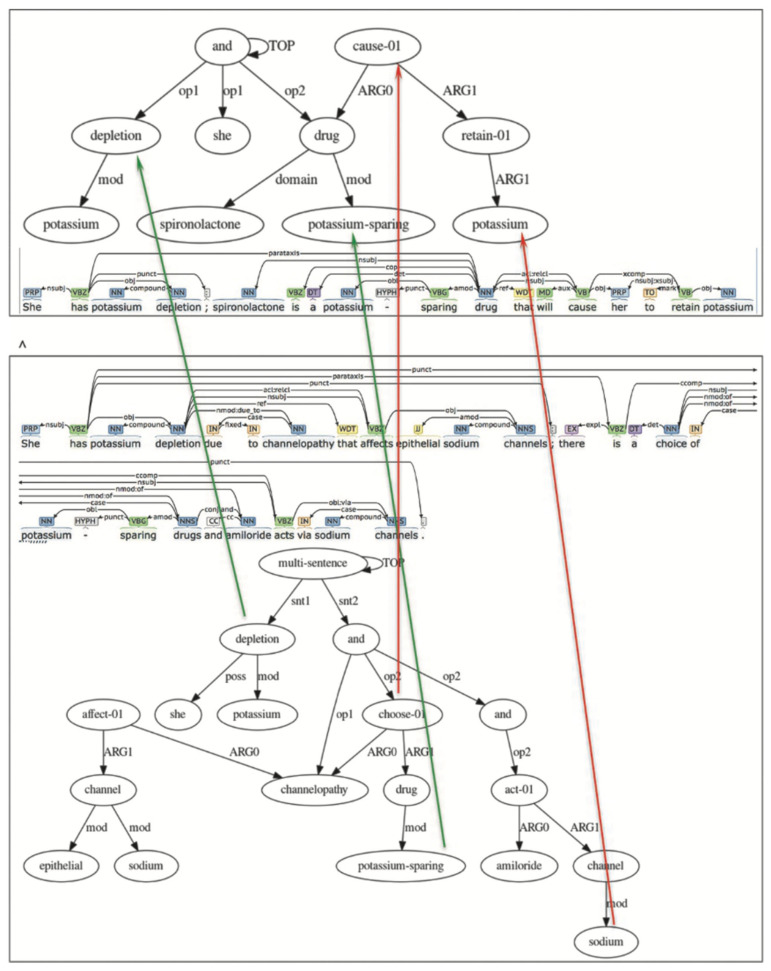
A map between semantic and syntactic representations for raw (incorrect, on the top) and the true (correct, on the bottom, generating low SEP) treatment of a disease. Green arrows show confirmation with the true content, and red arrows show rejection.

**Figure 4 entropy-25-01496-f004:**
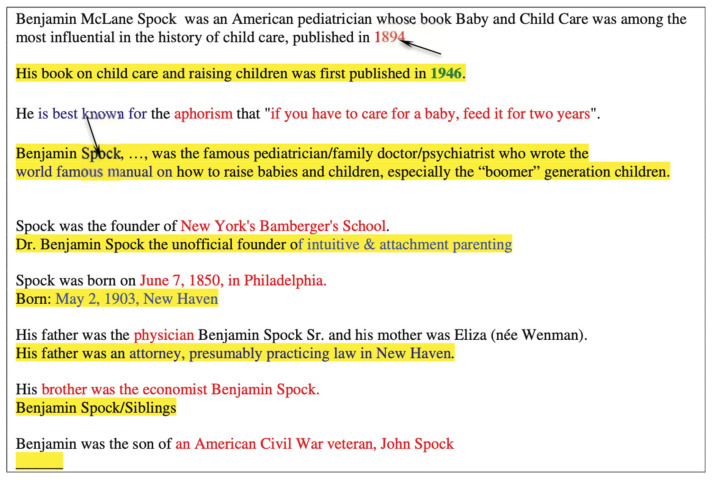
Fact-checking correction and extension of the raw content. Arrows show the mapping between hallucinating phrases/values and correct ones.

**Figure 5 entropy-25-01496-f005:**
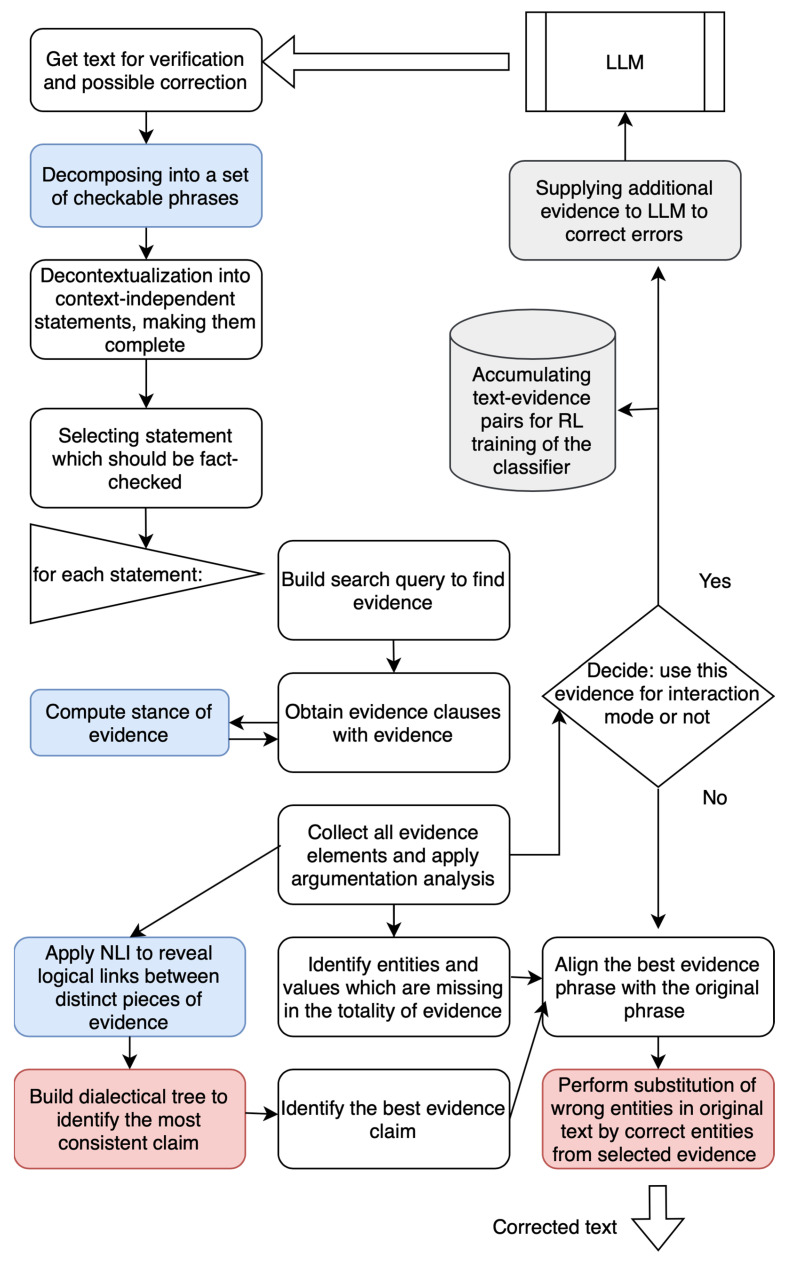
Content verification and correction architecture.

**Table 2 entropy-25-01496-t002:** Performance of a hybrid DNN + AMRG system on SQuAD 2.0.

	P	R	F1	kNN Entropy Estimate
AMR only	83.3	84.1	83.7 ± 0.4	134%
DNN ALBERT only			90.9 ± 0.3	117%
DNN + AMRG correcting DNN errors	92.7	92.3	92.6 ± 0.4	100%
DNN versus AMRG as selected by meta-learning	83.1	85.0	84.0 ± 0.3	115%
AMRG errors identification	76.8	77.2	77.0 ± 0.3	
AMRG errors correction	70.2	72.1	71.1 ± 0.4	

**Table 3 entropy-25-01496-t003:** Full-index search accuracy on SQuaD 2.0.

	P	R	F1	NDCG @ 5 Hits
Standalone AMRG	61.1	56.3	58.6 ± 0.3	0.331 ± 0.016
Standalone end-to-end DNN	64.7	62.4	63.5 ± 0.3	0.358 ± 0.012
DNN + AMRG correcting DNNs errors	66.2	63.3	64.7 ± 0.3	0.410 ± 0.015
DNN versus AMRG as selected by meta-agent	63.0	65.4	64.2 ± 0.3	0.342 ± 0.015

**Table 4 entropy-25-01496-t004:** Improvement over the abstractive summarization baseline.

	ROUGE-1	ROUGE-2	ROUGE-L	Improvement ROUGE-L, %	Corpus	kNN Entropy Drop
RASG [30]	30.33	12.39	27.16 ± 0.8	3.2 ± 0.3	Weibo	12.1%
CATTS [29]	44.9	22.6	37.3 ± 0.5	0.7 ± 0.3	SciTLDR	3.2%
LC GOLC [31]	29.38	10.38	27.18 ± 0.6	3.9 ± 0.4	CNN	14.0%
HATS [28]	42.16	19.17	38.35 ± 0.6	3.5 ± 0.3	CNN	13.4%

**Table 5 entropy-25-01496-t005:** (**a**) Characterizing the correction scenarios for GPT3. By reducing structural entropy costs (SEP), the outcome productivity entropy (MEP) increased. (**b**) Characterizing the correction scenarios for GPT4.

**(a)**										
**Class of Common Symptoms**	**#/% of Corrected Sentences (by Substituting Entities)**		**#/% of Corrected Sentences (by Substituting Phrases)**		**#/% of Rejected Sentences**		**#/% of Sentences Accepted as They Are**	**Total # of Raw Sentences**		**Total # of True Sentences Used**
Bloating	3.6	39.56	1.8	19.78	2.4	26.37	1.3	14.29	9.1	17.1
Cough	2.7	32.14	2.1	25.00	1.9	22.62	1.7	20.24	8.4	16.3
Diarrhea	3.0	33.71	2.1	23.60	2.3	25.84	1.5	16.85	8.9	19.0
Dizziness	2.7	30.68	2.3	26.14	2.5	28.41	1.3	14.77	8.8	18.7
Fatigue	2.9	35.80	1.9	23.46	2.0	24.69	1.3	16.05	8.1	20.4
Fever	3.5	36.84	2.0	21.05	2.4	25.26	1.6	16.84	9.5	19.3
Headache	3.0	36.59	1.7	20.73	2.1	25.61	1.4	17.07	8.2	17.5
Muscle Cramp	2.8	32.94	2.2	25.88	2.0	23.53	1.5	17.65	8.5	19.0
Nausea	3.2	36.78	2.0	22.99	2.3	26.44	1.2	13.79	8.7	17.9
Throat irritation	3.1	37.35	1.7	20.48	1.9	22.89	1.6	19.28	8.3	16.4
**Average**	**3.05**	**35.24**	**1.98**	**22.91**	**2.18**	**25.17**	**1.44**	**16.68**	**8.65**	**18.16**
**(b)**										
**Class of Common Symptoms**	**#/% of Corrected Sentences (by Substituting Entities)**		**#/% of Corrected Sentences (by Substituting Phrases)**		**#/% of Rejected Sentences**		**#/% of Sentences Accepted as They Are**	**Total # of Raw Sentences**		**Total # of True Sentences Used**
Bloating	1.58	17.76	0.80	8.83	1.68	19.71	2.41	25.16	12.3	15.83
Cough	1.16	14.45	0.89	11.20	1.28	16.85	2.98	35.55	10.3	15.57
Diarrhea	1.30	15.12	0.85	10.58	1.62	19.31	2.70	29.50	8.5	18.51
Dizziness	1.20	13.80	0.95	11.75	1.76	21.29	2.28	25.96	10.0	17.62
Fatigue	1.25	16.02	0.76	10.51	1.36	18.51	2.37	28.15	12.2	19.63
Fever	1.56	16.54	0.81	9.37	1.72	18.86	2.94	29.61	13.5	18.82
Headache	1.25	16.45	0.70	9.23	1.45	19.19	2.55	29.98	10.7	16.66
Muscle Cramp	1.25	14.73	0.94	11.60	1.37	17.55	2.73	30.98	12.4	18.86
Nausea	1.44	16.50	0.88	10.26	1.65	19.71	2.21	24.16	11.3	16.85
Throat irritation	1.36	16.78	0.72	9.20	1.36	17.04	2.93	33.88	10.0	14.45
**Average**	**1.34**	**15.82**	**0.83**	**10.26**	**1.53**	**18.80**	**2.61**	**29.29**	**11.1**	**17.28**

**Table 6 entropy-25-01496-t006:** The performance of repaired generated content (%).

The Class of Common Symptoms	Rate of Wrong Facts Per Text for Raw Content	Rate of Wrong Facts Per Text for Corrected Content	Rate of Wrong Facts Per Sentence for Raw Content	Rate of Wrong Facts Per Sentence for Corrected Content	Rate of Discourse Distortion Per Sentence
Bloating	89.6	15.3	68.4	6.4	23.1
Cough	93.5	16.4	70.2	7.1	21.8
Diarrhea	96.1	16.0	69.9	5.8	18.4
Dizziness	94.0	16.8	69.0	6.2	23.6
Fatigue	94.3	15.2	71.4	7.0	17.9
Fever	95.0	13.8	67.7	5.9	20.2
Headache	96.7	17.1	73.8	7.3	19.0
Muscle Cramp	94.2	15.6	68.2	6.7	17.3
Nausea	97.1	17.0	67.3	7.2	21.0
Throat irritation	95.3	15.9	69.0	6.4	19.8
**Average**	94.58	15.91	69.49	6.60	20.21

**Table 7 entropy-25-01496-t007:** Manual assessment of the error rate and ablation study (%).

System Architecture	Rate of Wrong Facts Per Text for Raw Content	Rate of Wrong Facts Per Text for Corrected Content	Rate of Wrong Facts Per Sentence for Raw Content	Rate of Wrong Facts Per Sentence for Corrected Content	Rate of Discourse Distortion Per Sentence
Complete system—auto assessment	94.6	15.9	69.5	6.6	20.2
Complete system—hybrid/manual assessment		17.5		7.2	22.0
Reduced web mining to a number of medical domains		17.1		11.3	26.3
Keyword-based generalization		23.4		10.7	25.6
Always do phrase substitution		18.7		9.5	25.4
Always do entity substitution		17.0		8.3	27.9

## Data Availability

Not applicable.

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
