# Peer review of "Applications of Shaped-Charge Learning"

_entropy, 2023, doi:10.3390/e25111496_

Round 1

Reviewer 1 Report

Technical refinement of the manuscript is required. For instance, Section I (Introduction) has only one subsection (1.1), prompting questions about its proper introduction. Within this section, a more comprehensive elucidation of the fundamental contributions made by the conducted research is essential, along with a clear distinction from the existing manuscript already accepted in this Special issue.

The Introduction section should also outline and acquaint the reader with the overall manuscript's structure.

Numerous self-references akin to those found in the previously accepted paper by the same author in this Special issue are present. Furthermore, several references such as 51, 47, 44, 28, 30, etc., are reiterated.

A revision of Figure 2 is necessary to enhance its readability.

The rationale behind selecting the ALBERT transformer model requires a more elaborate exposition.

In conclusion, a more detailed explication of the limitations inherent in the conducted research is imperative.

The author's specific contributions are not distinctly underscored in the provided text. This pertains to innovations within the implemented algorithm or advancements in developing a new dataset that could be valuable for subsequent research. A lucid articulation and thorough analysis of these contributions are indispensable.

Moderate editing of English language required

Author Response

Dear Reviewer:  thank you for review, I addressed all your comments

Reviewer 2 Report

The author proposes a methodology for applications in a text domain, where it is implemented meta-learning in the form of textual discourse. Syntax, semantics together with a deep learning recognition of syntax and semantics, while address forms the meta-level, controlling the object level and its performance.

The manuscript presents an exciting and effective methodology for question-answering and content-creation tasks. The results demonstrate the methodology's effectiveness; however, the text could be easier to read and follow. It contains much information, and there is information that is optional to the article. I would like the text to be more straightforward in its writing and that it could be easier to follow the information provided.

Some other minor comments:

a)      What is the meaning of ML and MEP abbreviations used in the abstract section? (Define the meaning of abbreviations in their first use.)

b)     Enumerations of the equations used in the text is required.

Author Response

(The authors gave the same response as above.)

Reviewer 3 Report

This paper proposes a new DNN architecture that is claimed to significantly improve the overall accuracy and explainability of ML, as well as  offering human-machine team support. This material is based on prior publications, e.g., [Galitsky 2023].

This paper is very hard to understand. First, the objectives are unclear, and the correspondence between the stated objectives and the paper's content is confusing.

"We evaluate the proposed architecture for question answering and content creation tasks and observe a significant improvement in performance along with enhanced usability by team members."

The actual content is:

3. Text summarization

4. Open domain text generation

5. Functioning in a team

It seems like "Text summarization" is different to "question answering", so the authors need to be precise about exactly what is claimed.

The article is rambling and unfocused. Sections 5 and 6 seem to play no role in the stated tasks of question answering and content creation tasks, and should be deleted.

What is the key innovation beyond [Galitsky 2023]? It seems like this is primarily an empirical validation. This could be stated very precisely instead of as a rambling narrative.

The following claim is unsupported:

"In this article, for applications in a text domain, we implement meta-learning in the form of textual discourse. Syntax, semantics together with a DNN’s recognition of syntax and semantics constitute the object level (i.e. the structure), while discourse forms the meta-level, controlling the object level and its performance (MEP)."

The main claim (l. 715) does not seem properly supported.

"Finally, extending a DNN with a kNN and performing meta-learning over this extension significantly improves the overall accuracy and explainability of ML as well as  offering human-machine team support, as was demonstrated in question-answering,  summarization and content generation work alone."

- the term "significantly improves" needs to be scientific. I want to see statistical significance measures.

- the results from Tables 1-3 are slight improvement, if any.

- the results from Table 4 are small improvement, so need a statistical significance measure.

The most interesting results are in Tables 5-7, so more time and focus should be placed on these experiments.

The claim (line 15) "In terms of entropy, we claim that this architecture simulates low structural entropy production, allowing the ML-team interaction structure to reach MEP for its performance." is unsupported and should either be empirically justified or removed.

The article makes claims about meta-learning, but it is *never* defined.

The authors treat the Shaped-charge learning architecture as a black box. 

However, the article claims (l. 49) to show the impact of regions of the archtecture: "When making a prediction,  we first implement neural gradual descent and then employ nearest neighbors on the resulting candidate prediction, minimizing SEP, to yield a precise and comprehensible outcome, performing MEP of the output."

- with such stong claims, it MUST be the case that the authors test each stage of the pipeline independently (DNN, kNN, MEP), to show the relative importance.

A better empirical study would look at the relative role of the DNN and kNN regions. A standard approach in ML is to use ablation studies for this purpose. This article just does a top-level empirical analysis and does not shed any light into *why* the proposed approach works, or the importance of the underlying architecture.

Given the current results, the authors cannot claim that there is understandability of the underlying architectures. The experiments only show very coarse outcomes.

Where does this claim come from (line 163)?:  "DNN-based approaches often fail due to a lack of a online phrase structure similar  to the one available in the training set, or an inability to generalize to cover various cases."

-- chatGPT seems to do this well....

Section by Section comments:

=====================

Sec 4. Open domain text generation (l. 384)

The main goal of our work is to combine meaningless raw content generated by a DNN-based system with factual content taken from various sources in order to create a meaningful and truthful open-domain text.

--confusing objective. Do you really intend to take a DNN-generated response that may not be fully understandable and improve its understandability? In what context is this a real-world task?

Sec. 5. "Functioning in a team" is a short and poorly-written section. It does not fit into the article and should be deleted. This is just a hand-waving argument and is not of any scientific merit in comparison to the prior experimental validation.

Sec. 6. "Maximum Entropy principle for Shaped Charge" also seems artificially tacked onto the end of the article. It seems to play no role in the core contributions and also should be deleted.

Sec 7. "Discussions and Conclusion" is rambling and unfocused. The discussion (lines 683-714) is odd, and does not follow from the experimental analysis. Unless it is clearly linked to the technical contributions it should be deleted.

The claim (l. 727) is unsupported: "While reinforcement learning does not guarantee 100% factual accuracy and does not provide attribution for verified and corrected facts, shaped charge leverages these features. That is, reducing the errors with lowered structural entropy costs  lead to a better output, characterized by maximum entropy production"

- the mathematical presentation of Sec. 6 is incomplete and does not support this claim. The authors make a much weaker claim "shaped charge learning can be characterized by this generalization of entropy towards non-ergodic processes" which is also not fully sound.

Minor points

================

Abstract: define kNN  (never defined in the entire paper)

Would prefer lines 44-71 to be deleted. This seems like speculation. DNN architectures are mathematical objects and if you can't define the mathematical properties I see no reason to have this speculation. The article [Galitsky 2023] has a proper scientific description---why have this speculation?

[Galitsky 2023] Galitsky B, Ilvovsky D, Goldberg S. Shaped-Charge Learning Architecture for the Human-Machine Teams. Entropy 2023

The English is good overall. The underlying technical structure of the article needs significant work.

Author Response

(The authors gave the same response as above.)

Round 2

Reviewer 1 Report

The manuscript in question extensively draws upon previously published materials. However, the distinct novel contribution made by the author, compared to their prior endeavors and in relation to the contributions of other researchers, lacks clarity in its presentation.

Minor editing of English language required

Author Response

Response to Reviewers

Reviewer’s concerns are shown in red, author’s reply – in green, and inserted text – in black

Reviewer 1

The manuscript in question extensively draws upon previously published materials. However, the distinct novel contribution made by the author, compared to their prior endeavors and in relation to the contributions of other researchers, lacks clarity in its presentation.

We now extended the Contribution part with two more paragraphs explaining kNN and meta-learning in the application domains.

Comments and Suggestions for Authors

 Thanks are due to the author for the revisions introduced. As described below, we still assess the paper to have significant technical problems that need to be addressed to enable the paper to be recommended for publication. These issues are highlighted below.

  1. The article remains rambling and unfocused. It is now longer than before, rather than more concise and focused.

It is now reduced in size

  1. Sections 5 and 6 still play limited role in the stated tasks of question answering and content creation tasks, and should be deleted.

deleted

  1. The author has not taken on the (hard) task of rewriting the article as required by the original review. Instead there are local patches and the article has increased in size. Many pieces of the article still do not fit together, and significant rewriting is necessary to make them fit together.

The paper is now subject to significant reduction so the remaining parts are expected to fit together well

  1. c.     The article remains difficult to read. It is improved, but only in localized parts. Overall the same level of unintelligibility remains due to lack of global revision. For example, the first paragraph of sec. 7. (Discussions and Conclusion)seems to be drawn from an entirely different article. The author claims “Our approach was to connect a kNN with a DNN through gradient descent inspired by the path kernel framework (Domingos 2020).” This claim occurs for the first time here, and the reference (Domingos 2020) appears nowhere else in the article. It seems like the authors must explain this at the beginning of the article.

This paragraph is removed

The discussion on use of kNN (sec. 5) seems entirely different to this claim. How does this sec. 7 and the sec. 5 algorithm relate?

As we removed the above paragraph, the contradiction does not exist any more

  1. The author has addressed the prior issues with incorrect claims, but several incorrect claims still remain. For example, see item (2) below.

  1. The claim in line 957 “shaped-charge can potentially deliver 100% correct content and backs it up.” is NOT justified in the article and all material about this must either be empirically validated or removed. Unless there is a clear theoretical proof of this in a general case, it appears to be false. Please see the later question on scalability of this approach.

removed

  1. Why has the author failed to remove the non-scientific discussion of shaped-charge projectile as requested in the initial round?
  2. Figure 1 and lines 64-75 of revision

removed

  1. Section 5 remains an outlier. "Functioning in a team" now is a long and poorly-written section. There is no empirical validation, and it remains discursive and unrelated to the rest of the material. It needs to be empirically validated or deleted!

removed

  1. Sections 1.1 and 1.2 should be put in a section of their own, titled “Preliminaries” or “Technical Background”. In the revised version it is unclear what role is played by “1.1 Nearest Neighbor Based Entropy Estimator”. The same is true for 1.2.

We restructured section numbering accordingly. We now explain the role of each subsection

TOC is now:

We put all entropy – related discussion in this new Section 2’

  1. 6.     What is the relationship between MEPr and MEP? There are no precise definitions for either.

Since we removed Section 6 we don’t need MEPr any more. MEP is now formally defined

  1. Section 6 has many undefined items, so it is impossible to follow or to check its correctness. There is no citation for Hanel et al (2014). You introduce the function f(N) with no definitions; same with G(k|Q). Even with a great deal of effort it is impossible to understand this section, or to know how it is related to the rest of the article.

Now removed

  1. 8.     I still have problems understanding sec. 4. “Open domain text generation”. The authors have added some important text, in particular:  “The main goal of this section is to identify hallucinations in a content generated by a DNN-based system and repair these hallucinations with factual content taken from various sources.” The issue is: how this will work in general open domain contexts? The examples are from very limited domains (e.g., medicine) where the kNN has a target or disease/drug pairs. In general text, it will be unclear what the target structure for the kNN is, and the authors provide no means to understand how this will be done.

kNN operates on explicit structures like syntactic and semantic trees. We match one word of the certain part of speech with another word, which can be synonymous or not synonymous. We also match phrases of the same type. At the same time, attention mechanism matches embeddings of phrases in a way hardly interpretable by a human.

Also, kNN for correction of content is domain independent.

 Clearly Transformers have attention methods to identify and rank relationships. But a kNN does not, and the space of possible clusters in open text is infinite.

We now started provided explanation for this in the Introduction – contribution part. We explain that kNN operates on explicit linguistic structures, interpretable by humans, rather than embeddings. Therefore, humans can perform feature engineering for kNN

  1. a.     So, how does the kNN generation work for open domain text generation, and how can you guarantee scalability? Without some attention mechanism it seems like this generalized approach is not possible.

It is a syntactic and semantic match based on domain-independent syntactic and semantic parsing. These parsers are also DNN-based these days. So, both DNN- attention mechanism and DNN parsing mechanism + graph match algorithms are domain-independent, they do not require ontologies. We explain it now in the beginning of the section.

kNN operates on a structural representation of sentences, unlike the DNN attention mechanism. kNN takes a syntactic and semantic representation of a generated sentence and matches it via graph mapping algorithms to syntactic and semantic representation of a true sentence identified by meta-learning on the trusted websites or other authoritative sources of information. This procedure will be described in detail in this section. Hence the correction of DNN content is domain- and language – independent if syntactic, semantic and web search components can support a given language.

  1. The authors claim that “factual content (is) taken from various sources.” How do you identify those sources? How do you identify the “correct” sources? This is never discussed. In open domain text, there is a huge space of sources to consider, and there is no principled method for selecting the “correct” sources.

At the time of writing, hallucination of most LLMs is much higher than factual inaccuracy of popular web resources, indexed by major search engines. Although the latter might have insignificant factual inaccuracies, they are may be 1000 times less frequent then hallucinations of LLMs.

For our evaluation, we select the web sites indexed and rated highly by major web search engines as a source of true information to perform our fact-checking of DNN generated content. Although factual errors are still possible, they occur about a thousand times less frequently than hallucinations in DNN content. Misinformation in the popular portals indexed by Google and Bing and shown in top results is very rare. Only in social media misinformation created by humans and hallucination rates are comparable (Suarez-Lledo and Alvarez-Galvez 2021). The reader can get an estimate for DNN hallucination rate from Table 6.

In conclusion, this article has lots of interesting ideas but fails in its objective to empirically validate the proposed architecture for summarization, question answering and content creation tasks. For summarization the objective is achieved, but for all others the paper fails. Further, the section on MEP is mathematically unintelligible, and one key reference, Hanel et al (2014), is not provided to allow anyone to check its veracity.

This section is removed

As a consequence, even after one revision we deem the paper to be unacceptable in its current revised form. It needs significant rewriting, and not just the cosmetic extensions that the author introduced to the initial draft.

The paper has been significantly rewritten

Reviewer 3 Report

 Thanks are due to the author for the revisions introduced. As described below, we still assess the paper to have significant technical problems that need to be addressed to enable the paper to be recommended for publication. These issues are highlighted below.

1.     The article remains rambling and unfocused. It is now longer than before, rather than more concise and focused.

a.     Sections 5 and 6 still play limited role in the stated tasks of question answering and content creation tasks, and should be deleted.

b.     The author has not taken on the (hard) task of rewriting the article as required by the original review. Instead there are local patches and the article has increased in size. Many pieces of the article still do not fit together, and significant rewriting is necessary to make them fit together.

c.     The article remains difficult to read. It is improved, but only in localized parts. Overall the same level of unintelligibility remains due to lack of global revision. For example, the first paragraph of sec. 7. (Discussions and Conclusion) seems to be drawn from an entirely different article. The author claims “Our approach was to connect a kNN with a DNN through gradient descent inspired by the path kernel framework (Domingos 2020).” This claim occurs for the first time here, and the reference (Domingos 2020) appears nowhere else in the article. It seems like the authors must explain this at the beginning of the article. The discussion on use of kNN (sec. 5) seems entirely different to this claim. How does this sec. 7 and the sec. 5 algorithm relate?

a.     The author has addressed the prior issues with incorrect claims, but several incorrect claims still remain. For example, see item (2) below.

2.     The claim in line 957 “shaped-charge can potentially deliver 100% correct content and backs it up.” is NOT justified in the article and all material about this must either be empirically validated or removed. Unless there is a clear theoretical proof of this in a general case, it appears to be false. Please see the later question on scalability of this approach.

3.     Why has the author failed to remove the non-scientific discussion of shaped-charge projectile as requested in the initial round?

a.     Figure 1 and lines 64-75 of revision

4.     Section 5 remains an outlier. "Functioning in a team" now is a long and poorly-written section. There is no empirical validation, and it remains discursive and unrelated to the rest of the material. It needs to be empirically validated or deleted!

5.     Sections 1.1 and 1.2 should be put in a section of their own, titled “Preliminaries” or “Technical Background”. In the revised version it is unclear what role is played by “1.1 Nearest Neighbor Based Entropy Estimator”. The same is true for 1.2.

6.     What is the relationship between MEPr and MEP? There are no precise definitions for either.

7.     Section 6 has many undefined items, so it is impossible to follow or to check its correctness. There is no citation for Hanel et al (2014). You introduce the function f(N) with no definitions; same with G(k|Q). Even with a great deal of effort it is impossible to understand this section, or to know how it is related to the rest of the article.

8.     I still have problems understanding sec. 4. “Open domain text generation”. The authors have added some important text, in particular:  The main goal of this section is to identify hallucinations in a content generated by a DNN-based system and repair these hallucinations with factual content taken from various sources.” The issue is: how this will work in general open domain contexts? The examples are from very limited domains (e.g., medicine) where the kNN has a target or disease/drug pairs. In general text, it will be unclear what the target structure for the kNN is, and the authors provide no means to understand how this will be done. Clearly Transformers have attention methods to identify and rank relationships. But a kNN does not, and the space of possible clusters in open text is infinite.

a.     So, how does the kNN generation work for open domain text generation, and how can you guarantee scalability? Without some attention mechanism it seems like this generalized approach is not possible.

b.     The authors claim that “factual content (is) taken from various sources.” How do you identify those sources? How do you identify the “correct” sources? This is never discussed. In open domain text, there is a huge space of sources to consider, and there is no principled method for selecting the “correct” sources.

In conclusion, this article has lots of interesting ideas but fails in its objective to empirically validate the proposed architecture for summarization, question answering and content creation tasks. For summarization the objective is achieved, but for all others the paper fails. Further, the section on MEP is mathematically unintelligible, and one key reference, Hanel et al (2014), is not provided to allow anyone to check its veracity.

As a consequence, even after one revision we deem the paper to be unacceptable in its current revised form. It needs significant rewriting, and not just the cosmetic extensions that the author introduced to the initial draft.

The English is basically good, but the new additions need some work.

Author Response

(The authors gave the same response as above.)

Round 3

Reviewer 3 Report

This article has improved considerably.  It is becoming a good scientific work with the changes.

The author still is making unsupported claims in the Conclusion that must be deleted or clarified.

1. What is the basis for these claims?

    ---"limited nature of language and its connection to causality": language and causality are powerful terms and their relations are not addressed. To my knowledge there is no notion of causality in language that is accepted by all. Delete this or make it precise.

    ---"However, once the assumptions about the connection between thought and language is dismissed, it is clear that LMs are stuck with a shallow understanding that will never reach the full-bodied thinking of humans." Delete this or make it precise.

2. line 721: "maintaining low structural complexity": This is never demonstrated in the article.

As requested in previous reviews, you MUST deal with the complexity of your approach if you claim it is a lower-complexity solution.

You raise the issue of finetuning of LLM parameters: "This can be exceedingly costly "(line 740)

What precisely is the complexity of your approach, to do a fair comparison?

3. You never properly answered the question from earlier reviews: "So, how does the kNN generation work for open domain text generation, and how can you guarantee scalability? Without some attention mechanism it seems like this generalized approach is not possible."

The success of your approach REQUIRES syntactic and semantic representations of generated sentences: it is expensive to create these.
You use a small set of such representations: what is needed to make this a general approach?

You cannot claim that your approach is general unless you explicitly answer these questions.
Otherwise, you must state that this method is limited to domains for which pre-computed syntactic and semantic representations of  sentences exist, and describe the limitations imposed.

The English is good. There are minor issues so a review prior to publishing would help.

--line 76---insert ):

(syntactic and semantic between two sentences.

--there are many situations where articles must be put in.

Author Response

Dear reviewer, I submitted my responses in the attached doc
